# Temporal Trends in the Prevalence, Treatment and Outcomes of Patients with Acute Coronary Syndrome at High Bleeding Risk

**DOI:** 10.3390/diagnostics12081784

**Published:** 2022-07-22

**Authors:** Ziad Arow, Tal Ovdat, Mustafa Gabarin, Alexander Omelchenko, Mony Shuvy, Tsafrir Or, Abid Assali, David Pereg

**Affiliations:** 1Cardiology Department, Meir Medical Center, Kfar Saba 4428164, Israel; ziad.arow@gmail.com (Z.A.); mustafa.gabarin88@gmail.com (M.G.); alexom36@gmail.com (A.O.); aassali@clalit.org.il (A.A.); 2Sackler Faculty of Medicine, Tel-Aviv University, Tel-Aviv 6997801, Israel; 3Department of Cardiology, Sheba Medical Center, Ramat Gan 8452566, Israel; tal.cohen@sheba.health.gov.il; 4Jesselson Integrated Heart Center, Shaare Zedek Medical Center, The Hebrew University, Jerusalem 9112102, Israel; monysh@gmail.com; 5Department of Cardiology, Galilee Medical Center, Nahariyya 2220903, Israel; tsafriro13@gmail.com; 6Azrieli Faculty of Medicine in the Galilee, Bar-Ilan University, Safed 5290002, Israel

**Keywords:** acute coronary syndrome, bleeding risk, CRUSADE Score

## Abstract

(1) Background: High bleeding risk is associated with adverse outcomes in ACS patients. We aimed to evaluate temporal trends in treatment and outcomes of ACS patients according to bleeding risk. (2) Methods: Included were ACS patients enrolled in ACSIS surveys. Patients were divided into three groups according to enrolment period: early (2002–2004), mid (2006–2010) and recent (2012–2018). Each group was further stratified into three subgroups according to CRUSADE bleeding risk score. The primary endpoints were 30-day MACE and 1-year all-cause mortality. (3) Results: Included were 13,058 ACS patients. High bleeding risk patients were less frequently treated with guideline-based medications and coronary revascularization. They also had higher rates of 30-day MACE and 1-year all-cause mortality regardless of the enrollment period. Among patients enrolled in early period, 30-day MACE rates were 10.8%, 17.5% and 24.3% (*p* < 0.001) and 1-year all-cause mortality rates were 2%, 7.7% and 23.6% (*p* < 0.001) in the low, moderate and high bleeding risk groups, respectively. Among patients enrolled in mid period, 30-day MACE rates were 7.7%, 13.4% and 23.5% (*p* < 0.001) and 1-year all-cause mortality rates were 1.5%, 7.2% and 22.1% (*p* < 0.001) in low, moderate and high bleeding risk groups, respectively. For patients enrolled in recent period, 30-day MACE rates were 5.7%, 8.6% and 16.2%, (*p* < 0.001) and 1-year all-cause mortality rates were 2.1%, 6% and 22.4%, (*p* < 0.001) in low, moderate and high bleeding risk groups, respectively. These differences remained significant following a multivariate analysis. (4) Conclusions: The percentage of patients at high bleeding risk has decreased over the last years. Despite recent improvements in the treatment of ACS patients, high bleeding risk remains a strong predictor of adverse outcomes.

## 1. Introduction

High bleeding risk is associated with a higher rate of adverse outcomes including short- and long-term mortality in patients with ACS [1,2,3,4,5]. Nevertheless, despite their high cardiovascular risk, ACS patients at high bleeding risk are more commonly selected for conservative management rather than an invasive strategy and are less frequently treated with guideline-based medical therapy [1]. In recent decades, the treatment of patients with ACS has improved dramatically with the introduction of new anti-platelet agents, potent lipid-lowering medications and major advances in both percutaneous and surgical coronary revascularizations [6,7,8,9]. This improved treatment has been associated with a subsequent reduction in mortality and cardiovascular complications among different populations of ACS patients [10,11,12]. Whether the improvement in treatment has resulted in a better outcome for ACS patients at high bleeding risk has not been evaluated yet. The current study was aimed to evaluate temporal trends in the prevalence, treatment and outcomes of ACS patients at high bleeding risk.

## 2. Materials and Methods

***Study population:*** The Acute Coronary Syndromes Israeli Survey (ACSIS) is carried out for 2 months every 2–3 years in all intensive coronary care units and cardiology departments in Israel. The study population consisted of patients presenting with ST-elevation and non-ST-elevation myocardial infarction or unstable angina pectoris that were included in the ACSIS Surveys during 2000–2018. Study physicians recorded all clinical and demographic data on pre-specified forms for consecutive participants. The diagnosis of ACS was based on clinical, electrocardiographic and biochemical criteria, and patients were managed at the discretion of each medical center.

***Bleeding risk assessment:*** The CRUSADE bleeding risk score (heart rate, systolic blood pressure, hematocrit, creatinine clearance, sex, signs of heart failure at admission and history of vascular disease or diabetes mellitus) was used for bleeding risk assessment [13]. It was calculated for each individual patient at the presentation to the hospital and patients were stratified into three groups of low, intermediate and high bleeding risk (CRUSADE score 1–30, 31–40 and 41–96, respectively). We compared baseline characteristics, treatment and clinical outcomes of ACS patients according to bleeding risk in the entire ACSIS population and in a separate analysis for patients enrolled in early (2002, 2004), mid (2006, 2010) and late (2013, 2015, 2018) surveys.

***Outcomes:*** The primary endpoints of the study were 1-year all-cause mortality and 30-day major adverse cardiovascular events (MACE). MACE was the composite of all-cause mortality, myocardial infarction and cerebrovascular accident (CVA). Mortality rates were determined for all participants from hospital charts and by matching the identification numbers of the patients with the Israeli National Population Registry.

***Statistical analysis:*** Patients’ characteristics were presented as *n* (%) for categorical variables, and as mean (sd) or median (IQR) for normal/non-normal distributed continuous variables. The cohort was divided into three groups according to enrolment periods (early, mid and recent). Each group was further divided into three subgroups according to the CRUSADE bleeding risk score (low, intermediate and high risk). Baseline characteristics, treatment and clinical outcomes were compared between patients in the different bleeding risk groups within each time period. Chi-square test was used for comparison of categorical variables. Analysis of variance with one degree of freedom was performed for comparison of normally distributed continuous variables. The Kendall rank correlation was performed for non-normal distribution. Kaplan–Meier curves were used to present 1-year survival rates. The comparison of outcomes between different bleeding risk categories within the same enrolment period was performed using a pairwise log rank test with Holm’s *p*-value adjustment. Multivariate adjustment was further conducted using Cox proportional hazard models. Included in the adjustment were relevant baseline demographic and clinical characteristics not already included in the CRUSADE score. All analyses were performed using R (R-studio, V.4.0.3, Vienna, Austria).

## 3. Results

The 13,058 ACS patients had a median age of 63 years and included 77.9% men. Of them, 3702 were enrolled at the early period, 5248 at mid and 4108 at the recent periods. Each time period was further divided into three subgroups according to CRUSADE bleeding risk. The percentage of patients at high bleeding risk decreased in recent surveys. Baseline characteristics according to enrolment period and bleeding risk are presented in Table 1. At all enrolment periods, patients at high bleeding risk were older, more frequently women and more commonly presented with cardiovascular risk factors and a history of cardiovascular disease. Accordingly, patients with higher bleeding risk were more commonly treated with cardiovascular medication prior to hospitalization.

***In hospital treatment according to bleeding risk:*** In hospital treatment characteristics according to enrolment period and bleeding risk are presented in Table 2. Regardless of enrollment period, ACS patients at high bleeding risk were less frequently treated with guideline-based medical therapy, including anti-platelet agents, statins and ACE-inhibitors/ARBs. Among high bleeding risk patients enrolled in recent surveys, anti-platelet therapy less commonly included ticagrelor or prasugrel rather than clopidogrel. Moreover, referral rates for an invasive strategy with coronary angiography and subsequent coronary angioplasty during hospitalization were lower in the high bleeding risk groups regardless of enrolment period. Referral for surgical revascularization during hospital admission decreased in recent compared to early enrolment periods, regardless of bleeding risk.

***Outcome of ACS patients according to bleeding risk:*** Kaplan–Meier curves comparing 1-year mortality according to bleeding risk for each enrolment period are presented in Figure 1. High bleeding risk was associated with increased mortality rates regardless of enrolment period. We further conducted a multivariate analysis for 1-year all-cause mortality for the study groups. An analysis using the low bleeding risk as a reference demonstrated significantly higher 1-year mortality rates among patients at high bleeding risk in all enrolment periods. Among patients enrolled in early period, odds ratio for 1-year all-cause mortality were 2.88, 95%CI 1.8–4.4, *p* < 0.001 and 6.35, 95%CI 4.2–9.5, *p* < 0.001 in moderate and high bleeding risk compared to low bleeding risk group, respectively. For patients enrolled in mid period, odds ratio for 1-year all-cause mortality were 3.65, 95%CI 2.4–5.5, *p* < 0.001 and 8.6, 95%CI 5.8–12.7, *p* < 0.001 in moderate and high bleeding risk compared to low bleeding risk group, respectively. For patients enrolled in the recent period, the odds ratio for 1-year all-cause mortality was 2.2, 95%CI 1.4–3.5, *p* = 0.001 and 5.8, 95%CI 3.9–8.6, *p* < 0.001 in moderate and high bleeding risk compared to low bleeding risk group, respectively.

High bleeding risk was associated with increased risk of 30-day MACE regardless of enrolment period (Figure 2). We further conducted a multi-variant analysis for 30-day MACE for the study groups. An analysis using the low bleeding risk as a reference demonstrated significantly higher 30-day MACE rates among patients at high bleeding risk in all enrolment periods. Among patients enrolled in early period, the odds ratio for 30-day MACE was 1.45, 95%CI 1.1–1.8, *p* = 0.01 and 1.8, 95%CI 1.3–2.3, *p* < 0.001 in moderate and high bleeding risk compared to low bleeding risk group, respectively. For patients enrolled in mid period, the odds ratio for 30-day MACE was 1.48, 95%CI 1.1–1.9, *p* = 0.004 and 2.57, 95%CI 1.9–3.3, *p* < 0.001 in moderate and high bleeding risk compared to low bleeding risk group, respectively. For patients enrolled in the recent period, the odds ratio for 30-day MACE was 1.47, 95%CI 1.02–2.1, *p* = 0.04 and 2.38, 95%CI 1.6–3.3, *p* < 0.001 in moderate and high bleeding risk compared to low bleeding risk group, respectively.

## 4. Discussion

The current study demonstrated a significant improvement in the treatment of ACS patients over the past two decades. ACS patients included in recent surveys were more frequently selected for an invasive strategy with coronary angiography and subsequent revascularization and were more commonly treated with guideline-based medical therapy. Nevertheless, even in the recent period, high bleeding risk remained a strong predictor for adverse cardiovascular outcomes and mortality. Despite their increased cardiovascular risk, ACS patients at high bleeding risk were more commonly selected for conservative management rather than an invasive strategy and were less frequently treated with guideline-based medical therapy regardless of enrolment period.

Several risk scores have been suggested for the assessment of bleeding risk in patients with ACS including the CRUSADE, ACUITY and the Academic Research Consortium for High Bleeding Risk (ARC-HBR) bleeding risk score [1]. The CRUSADE bleeding score was initially developed for NSTE-ACS patients [13] and was subsequently also validated for patients with STEMI [14]. A meta-analysis comparing different bleeding scores performance including 18,155 ACS patients from 17 studies demonstrated that the CRUSADE score was the most widely used score, and also performed better especially in patients selected for invasive strategy [15]. These findings were further supported by additional studies [16,17]. Accordingly, the CRUSADE score has been recommended by international clinical guidelines for bleeding risk assessment in ACS [1].

The association between high bleeding risk and adverse outcomes including in-hospital and 1-year mortality of patients with ACS has been demonstrated in several studies [2,3,4,5]. However, the current study is, to the best of our knowledge, the first to assess temporal trends in the treatment and outcome of ACS patients according to their bleeding risk. The association between bleeding risk and outcome is not completely understood and appears to be multifactorial. First, both major and minor bleeding events have been well demonstrated to be associated with poor outcomes in ACS patients including short- and long-term mortality [18,19]. Second, concerns regarding major bleeding events may influence both patient’s and physician’s treatment decisions, including the preference of a non-invasive strategy in order to minimize the need for anticoagulation and potent anti-platelet therapy. Indeed, we found that ACS patients at high bleeding risk were more commonly selected for a conservative rather than an invasive strategy. Moreover, patients at high bleeding risk were less frequently treated with guideline-based medical therapy. While the lower rate of treatment with antiplatelet drugs may be explained by the increased bleeding risk, patients at high bleeding risk were also less frequently treated with statins and ACE inhibitors. Finally, the presence of high bleeding risk is also associated with an older age and baseline cardiovascular comorbidities including diabetes, heart failure, previous MI, stroke and chronic kidney disease, which are all independently associated with poor outcomes in patients with ACS.

In recent years several strategies to reduce bleeding in ACS patients have been developed, including the preference of the radial approach, avoidance of pre-loading with anti-platelet therapy, gastric protection with proton pump inhibitors and the option to shorten dual anti-platelet treatment duration in patients at high bleeding risk. These strategies have resulted in better treatment and outcomes in the general population of ACS patients. Indeed, we demonstrated higher rates of referral for an invasive strategy and treatment with guideline-based medical therapy in recent surveys regardless of bleeding risk. Nevertheless, even with contemporary therapy, patients at high bleeding risk remain highly susceptible to adverse cardiovascular outcomes.

Our study has several limitations that warrant consideration. First, our database did not include data regarding post-discharge bleeding events and the very low rates of in-hospital major bleeding events precluded a meaningful analysis. Second, since the specific cause of death was not available, the primary endpoint of our study was all-cause rather than cardiovascular mortality. Finally, the ACSIS is a large national survey and therefore our findings should be extrapolated to other countries with caution.

## 5. Conclusions

The percentage of patients at high bleeding risk decreased in recent surveys. Despite the improvement in the treatment of ACS patients in recent years, high bleeding risk remains a strong predictor of poor clinical outcomes.

## Figures and Tables

**Figure 1 diagnostics-12-01784-f001:**
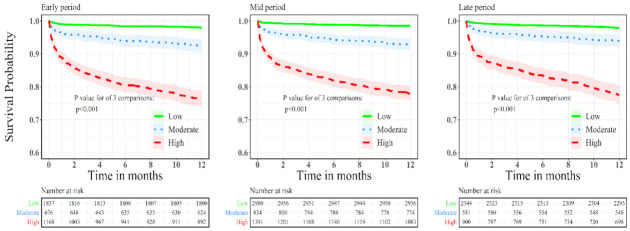
Kaplan–Meier curves for 1-year all-cause mortality according to bleeding risk for patients enrolled in early, mid and recent surveys.

**Figure 2 diagnostics-12-01784-f002:**
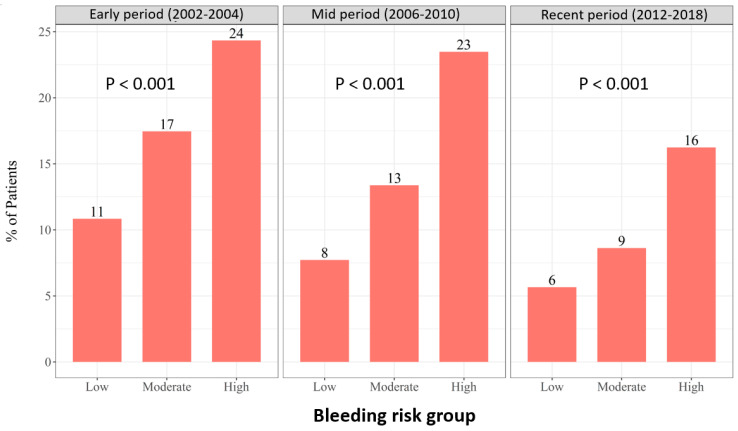
The 30-day MACE according to bleeding risk and enrolment period.

**Table 1 diagnostics-12-01784-t001:** Baseline characteristics according to bleeding risk.

Baseline Characteristics		CRUSADE Bleeding Risk		
	Low	Moderate	High	*p*-Value
*n*	7386	2147	3525	
Age, years (median)	57	69	75	<0.001
Gender, male *n* (%)	6748 (91)	1496 (70)	1927 (55)	<0.001
Dyslipidemia, *n* (%)	4746 (64)	1485 (69)	2462 (70)	<0.001
Hypertension, *n* (%)	3578 (48)	1477 (69)	2793 (79)	<0.001
Diabetes mellitus, *n* (%)	1745 (24)	900 (42)	2071 (59)	<0.001
CKD, *n* (%)	109 (1.5)	175 (8)	1157 (33)	<0.001
Prior MI, *n* (%)	1664 (23)	731 (34)	1612 (46)	<0.001
Prior CABG, *n* (%)	387 (5)	261 (12)	597 (17)	<0.001
Prior CVA/TIA, *n* (%)	260 (3.5)	183 (8.5)	619 (18)	<0.001
PVD, *n* (%)	215 (3)	200 (9)	652 (19)	<0.001
**Prior medications**				
Aspirin, *n* (%)	2709 (38)	1122 (54)	2158 (63)	<0.001
Clopidogrel, *n* (%)	462 (7)	207 (10)	468 (14)	<0.001
ACE-I/ARB, *n* (%)	2044 (28)	959 (45)	1862 (53)	<0.001
Beta blockers, *n* (%)	1908 (27)	883 (43)	1797 (53)	<0.001
Statins, *n* (%)	2677 (39)	1078 (53)	1926 (57)	<0.001

CKD = chronic kidney disease; MI = myocardial infarction; CABG = coronary artery bypass graft; CVA = cerebrovascular accident; TIA = transient ischemic attack; PVD = peripheral artery disease; ACE-I = angiotensin-converting enzyme inhibitor; ARB = angiotensin receptor blockers.

**Table 2 diagnostics-12-01784-t002:** In hospital treatment characteristics according to enrolment period and bleeding risk.

Bleeding Risk	Low	Mod	High	*p*-Value	Low	Mod	High	*p*-Value	Low	Mod	High	*p*-Value
*n*	1855	676	1171		3006	845	1397		2525	626	957	
**Reperfusion therapy, *n* (%)**												
Coronary angiography	1714 (92)	619 (92)	952 (81)	<0.001	2855 (95)	744 (88)	1012 (72)	<0.001	2446 (97)	593 (95)	773 (81)	<0.001
PCI	1160 (63)	362 (54)	455 (39)	<0.001	2324 (77)	555 (66)	690 (49)	<0.001	1915 (76)	438 (70)	535 (56)	<0.001
CABG	51 (7)	22 (10)	49 (12)	0.01	138 (5)	74 (9)	74 (5)	0.09	130 (5)	32 (5)	59 (6)	0.2
**Treatment at discharge, *n* (%)**												
Aspirin	1779 (96)	622 (92)	988 (84)	<0.001	2938 (98)	807 (97)	1214 (92)	<0.001	2461 (98)	599 (97)	818 (91)	<0.001
P2Y12 inhibitor	1245 (67)	398 (59)	499 (43)	<0.001	2580 (86)	637 (76)	922 (70)	<0.001	2309 (92)	558 (91)	762 (85)	<0.001
Clopidogrel	1245 (67)	398 (59)	499 (43)	<0.001	2579 (86)	633 (76)	921 (70)	<0.001	608 (24)	247 (40)	486 (54)	<0.001
Prasugrel	0	0	0	-	1 (0)	4 (0.5)	1 (0.1)	0.3	883 (35)	107 (17)	74 (8)	<0.001
Ticagrelor	0	0	0	-	0	0	0	-	818 (33)	204 (33)	202 (22)	<0.001
Statins	1483 (80)	509 (75)	736 (63)	<0.001	2879 (96)	784 (94)	1178 (89)	<0.001	2429 (98)	591 (98)	828 (94)	<0.001
Beta blockers	1534 (83)	554 (82)	811 (69)	<0.001	2489 (83)	685 (82)	1049 (79)	0.001	1921 (81)	499 (84)	698 (80)	0.52
ACEI/ARB	1255 (68)	489 (72)	825 (71)	0.07	2333 (78)	681 (82)	961 (73)	0.001	1998 (85)	510 (87)	624 (74)	<0.001
Hypoglycemic drugs	202 (11)	105 (16)	221 (19)	<0.001	402 (13)	196 (23)	338 (25)	<0.001	492 (20)	208 (33)	297 (31)	<0.001

PCI = percutaneous coronary intervention; CABG = coronary artery bypass graft; ACE-I = angiotensin-converting enzyme inhibitor; ARB = angiotensin receptor blockers.

## Data Availability

Not applicable.

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
