# Peer review of "Temporal Trends in the Prevalence, Treatment and Outcomes of Patients with Acute Coronary Syndrome at High Bleeding Risk"

_diagnostics, 2022, doi:10.3390/diagnostics12081784_

Round 1

Reviewer 1 Report

Interesting study! However even though statistics is well conducted I do believe in general more in visual graphs compared with numbers, but I guess that is a personal preference and not a mandatory condition.

Regarding scientific importance , the large database makes it an imposing study , with good registry management.

Author Response

We thank the reviewer  for the thorough review of our manuscript and for the support. 

Reviewer 2 Report

I read with interest the paper by Arow and colleagues. Authors should be congratulated for the manuscript. I have no comments.

Author Response

We thank the reviewer  for the thorough review of our manuscript and for the suupport. 

Reviewer 3 Report

The article presents the data on the influence of high bleeding risk on treatment and outcomes of acute coronary syndrome as well as the temporal trend of this influence. The authors concluded, that despite recent improvements in the treatment of ACS patients, high bleeding risk remains a strong predictor of adverse outcomes.

Major comment:

The title of the article, the purpose of the study in the abstract and in the introduction are not similar. Please, unify them. Did you investigate the prevalence of the high bleeding risk? If so, indicate it  in the aim stated in the abstract and add your conclusion on this issue in the Conslusion section. There is no conclusion on the prevalence of high bleeding risk among ACS patients and its temporal trend. 

Author Response

We thank the reviewer  for the thorough review of our manuscript . We completley agree with all comments and have revised the manuscript accordingly. 

1. We have unified the title of the article, the purpose of the study in the abstract and in the introduction.

2. We have added data regarding the prevalence of high bleeding risk among ACS patients to the conclusions (abstract and full manuscript)